# Bundled Causal History Interaction

**DOI:** 10.3390/e22030360

**Published:** 2020-03-20

**Authors:** Peishi Jiang, Praveen Kumar

**Affiliations:** Ven Te Chow Hydrosystem Laboratory, Civil and Environmental Engineering, University of Illinois, Urbana, IL 61801, USA; pjiang6@illinois.edu

**Keywords:** bundled causal dynamics, information measures, complex system

## Abstract

Complex systems arise as a result of the nonlinear interactions between components. In particular, the evolutionary dynamics of a multivariate system encodes the ways in which different variables interact with each other individually or in groups. One fundamental question that remains unanswered is: How do two non-overlapping multivariate subsets of variables interact to causally determine the outcome of a specific variable? Here, we provide an information-based approach to address this problem. We delineate the temporal interactions between the bundles in a probabilistic graphical model. The strength of the interactions, captured by partial information decomposition, then exposes complex behavior of dependencies and memory within the system. The proposed approach successfully illustrated complex dependence between cations and anions as determinants of *pH* in an observed stream chemistry system. In the studied catchment, the dynamics of *pH* is a result of both cations and anions through mainly synergistic effects of the two and their individual influences as well. This example demonstrates the potentially broad applicability of the approach, establishing the foundation to study the interaction between groups of variables in a range of complex systems.

## 1. Introduction

In complex systems shaped by the interaction of a multitude of variables, an interesting question that remains unanswered is: In what ways do the evolutionary history of two subsets of variables interactively influence the current state of a target variable? Answering this question would be extremely useful in furthering our understanding in the collective behavior of a system’s dynamics, where the interactions of variables in groups play a key role. For instance, in the study of connectivity between different regions of the brain, one may be interested in how a specific reaction pulse is jointly induced by different groups of incoming signals [1]. In stream chemistry, which is shaped by numerous biophysical processes and chemical reactions in both the stream and the contributing landscape, one may be interested in understanding how stream *pH* level is an outcome of the joint effect of the concentrations of different anions and cations [2]. Addressing how the state of a specific variable at any time is a causal outcome of interaction from the entire or a part of the evolutionary history of a system requires a quantitative approach.

At the most elementary level, this approach calls upon investigating whether and in what way two variables interact with each other (Figure 1a). This type of pairwise interaction has been widely addressed by causal influence analysis [3,4,5,6]. Generally, most causality detection and analysis fall into two categories: the intervention- and non-intervention-based approaches. While the interventional analysis (e.g., Pearl’s causality [4]) is more intuitive by investigating how intervening a source variable impacts the target variable, the non-interventional analysis is more applicable in most real life systems where artificial interventions are almost impossible in a multivariate situation (Figure 1b–d). Among the existing non-interventional approaches, Granger causality [3] has gained significant popularity due to its intuitive statistical interpretation using observed time series to evaluate the cause–effect relation between two variables. This is done by quantifying the reduced uncertainty of the target that is explained by the source conditioned on the knowledge of the remaining variables in the system. In this study, the causal analysis is referred in this Granger sense.

While the original Granger causality only assesses the change of the second-order moment to determine the reduced uncertainty, recent studies extend it to capture the change of the entire probability space by using information measures [7,8]. For example, transfer entropy [7] quantifies the shared dependency between the current state of a target and the previous state of a source variable given the knowledge of the previous state of the target. Besides, momentary information transfer [8] combines multivariate transfer entropy with probabilistic graphical model to efficiently estimate the information flow between two lagged variables in a multivariate system. Information measures, therefore, are the core utilities for quantitative analysis in this study, in that they are capable of delineating the nonlinear interactions.

Here, we propose new information measures to quantify and characterize the interactive strength among two bundled variable sets in affecting the present state of a target variable (Figure 1c). This approach allows us: (1) to consider the effect of the entire evolutionary history of all interacting variables, termed causal history [9,10], that determines the current state of a variable of interest; and (2) to characterize such effect by using partial information decomposition (PID) [11] framework. We aim to analyze the joint influence from the evolutionary history of two subsets of variables, thus requiring multivariate interaction assessment. This draws upon another key feature of information measure—characterizing the dynamics among multiple components. For instance, to investigate the total information from two source variables to a target (Figure 1b), PID is used for decomposing the total information into different information contents identified as unique, redundant and synergistic. In addition, we further the PID approach to characterize the information from two lagged sources through specific pathways by conditioning the remaining system’s dynamics that are not in the pathways of interest [9]. This approach is called momentary partial information decomposition, building on the idea of momentary information and PID. Another illustrative example of using information measures to assess multivariate interaction is the causal history analysis framework [10,12], which accounts for the influence from the entire evolutionary dynamics of the system (Figure 1c).

The main contribution of this study is that it is the first time to delineate the overall effect of multiple interacting grouped sources to a target using information theory. It should be noted that the proposed method is fundamentally different from existing information measures. Information theory has been extensively employed to assess pairwise interactions (e.g., transfer entropy [7] and momentary information transfer [8]), i.e., whether and how a source affects a target. Some recent efforts take one step further to unravel multivariate interactions (e.g., PID and the previous causal history analysis framework). Nevertheless, neither the pairwise interaction nor the current multivariate analysis using information theory takes into account group interactions, which is the key in this study. By grouping variables, the proposed analysis targets dynamics of specific sets of components, thus providing new insights on interactions among different subsets of a system. Such insights cannot be obtained using previous analysis treating variables as individual components.

The rest of the paper is organized as follows. Section 2 details the proposed information measures for analyzing the bundled causal history interaction. They are developed based on a directed acyclic graph (DAG) representation for time-series. The DAG further serves as the basis for dimensionality reduction of the measures to ensure reliable information estimations. In Section 3, the bundled causal history analysis is employed to investigate the joint influence on the stream *pH* from cations and anions by using a set of observed stream chemistry data. Last, a brief conclusion is drawn in Section 4.

## 2. Methodology

We consider complex systems that can be conceptualized as a multivariate system consisting of *N* variables ${\vec{X}}_{t}={\left\{X_{t}^{\left(1\right)},X_{t}^{\left(2\right)},X_{t}^{\left(3\right)},\ldots ,X_{t}^{\left(N\right)}\right\}}_{N}$, varying in time *t*. The current state of a target variable, $X_{t}^{\mathrm{tar}}\in {\vec{X}}_{t}$, is an outcome of interactions in the entire evolutionary dynamics in the causal history [10] prior to time *t*, ${\vec{X}}_{:t}=\left\{\ldots ,{\vec{X}}_{t-3},{\vec{X}}_{t-2},{\vec{X}}_{t-1}\right\}$. Among the influence from the entire causal history in ${\vec{X}}_{:t}$, here, we investigate the joint effect from the historical states of two specific groups of variables in ${\vec{X}}_{:t}$ on $X_{t}^{\mathrm{tar}}$. We denote a bundled set containing $N_{m}\left(<N\right)$ variables at time *t* as ${\vec{X}}_{t}^{m}\subset {\vec{X}}_{t}$. The entire historical states of ${\vec{X}}_{t}^{m}$ is represented as ${\vec{X}}_{:t}^{m}=\left\{\ldots ,{\vec{X}}_{t-3}^{m},{\vec{X}}_{t-2}^{m},{\vec{X}}_{t-1}^{m}\right\}$. Now, the aim of our study is to characterize how $X_{t}^{\mathrm{tar}}$ is jointly driven by causal histories of two bundled sets, ${\vec{X}}_{:t}^{m}$ and ${\vec{X}}_{:t}^{n}$, where ${\vec{X}}_{:t}^{m}\cap {\vec{X}}_{:t}^{n}=\emptyset$.

In the rest of this section, we first develop the information measures for delineating the information flow from the two bundled sets, ${\vec{X}}_{:t}^{m}$ and ${\vec{X}}_{:t}^{n}$, to the target $X_{t}^{\mathrm{tar}}$. Then, to achieve reliable information measure estimation, we introduce a two-stage dimensionality reduction approach to reduce the cardinality of the proposed measures.

### 2.1. Interactive Information Flow from Two Bundled Variables

Let us denote the remaining variables in the system outside of the two chosen bundles as ${\vec{X}}_{t}^{\mathrm{rest}}={\vec{X}}_{t}\backslash \left\{{\vec{X}}_{t}^{m},{\vec{X}}_{t}^{n}\right\}$ with \ as the exclusion symbol. The total information, $T_{mn}$, given by the evolutionary dynamics of two bundled sets to the target $X_{t}^{\mathrm{tar}}$ can then be expressed as a conditional mutual information (CMI) [13] between the two bundled sets given the knowledge of ${\vec{X}}_{:t}^{\mathrm{rest}}$, which is given by: (1)$$\begin{array}{c} \begin{array}{c} T_{mn}=I\left(X_{t}^{\mathrm{tar}};{\vec{X}}_{:t}^{mn}|{\vec{X}}_{:t}^{\mathrm{rest}}\right), \end{array} \end{array}$$
where ${\vec{X}}_{:t}^{mn}={\vec{X}}_{:t}^{m}\cup {\vec{X}}_{:t}^{n}$. The conditioning on ${\vec{X}}_{:t}^{\mathrm{rest}}$ is to exclude the influence from the interactions of the rest of the system in the quantification of the interaction between the two bundles. Note that $X_{t}^{\mathrm{tar}}$ can belong to any variable in one of the bundled sets or to that in ${\vec{X}}_{t}^{\mathrm{rest}}$. When $X_{t}^{\mathrm{tar}}\in {\vec{X}}_{t}^{\mathrm{rest}}$, Equation (Equation 1) can be considered as a generalized transfer entropy [7]. Transfer entropy captures the reduction in the uncertainty associated with the prediction of the current state of a variable given the knowledge of another variable that is in addition to that from the knowledge of its own history. This generalization allows us to characterize the reduction in uncertainty from multiple variables that are in addition to those provided by the variables own history or that of a set to which it belongs.

To characterize the information contents in $T_{mn}$, we take advantage of Partial Information Decomposition (PID) [11], which allows us to decompose $T_{mn}$ into: (1) redundant information $R^{T_{}}$—the overlapping information given by two bundled causal histories, ${\vec{X}}_{:t}^{m}$ and ${\vec{X}}_{:t}^{n}$; (2) synergistic information $S^{T_{}}$—the joint information given by ${\vec{X}}_{:t}^{m}$ and ${\vec{X}}_{:t}^{n}$; and (3) unique information $U_{m}^{T_{}}$ and $U_{n}^{T_{}}$—the information provided by each bundled set, ${\vec{X}}_{:t}^{m}$ and ${\vec{X}}_{:t}^{n}$, respectively. This is given by: (2)$$\begin{array}{c} \begin{array}{c} T_{mn}=R^{T_{}}+S^{T_{}}+U_{m}^{T_{}}+U_{n}^{T_{}}. \end{array} \end{array}$$

Before we develop quantitative estimation, we ask another related question: Do all the historical states in the bundled sets provide information to $X_{t}^{\mathrm{tar}}$? If the answer is no, then when does such influence end as the lag between $X_{t}^{\mathrm{tar}}$ and any source node in ${\vec{X}}_{:t}^{mn}$ increases? Otherwise, how much information is given by very early dynamics in ${\vec{X}}_{:t}^{mn}$? Answering these questions requires the assessment of the memory dependencies due to the bundled causal history ${\vec{X}}_{:t}^{mn}$ on $X_{t}^{\mathrm{tar}}$. Therefore, we partition the entire bundled causal history into two complementary components: (1) a recent dynamics from all the states up to a positive time lag $\tau_{C}$, ${\vec{X}}_{t-\tau_{C}:t}^{mn}=\left\{{\vec{X}}_{t-\tau_{C}}^{mn},\ldots ,{\vec{X}}_{t-1}^{mn}\right\}$, termed immediate bundled causal history; and (2) the remaining earlier dynamics, ${\vec{X}}_{:t-\tau_{C}}^{mn}$, termed distant bundled causal history. By using the chain rule of CMI [14], we can decompose $T_{mn}$ in Equation (Equation 1) into the information from the immediate ($J_{mn}$) and distant ($D_{mn}$) bundled causal histories, which are given by: (3a)$$\begin{array}{cc} T_{mn}= & \underset{=J_{mn}}{\underset{︸}{I(X_{t}^{\mathrm{tar}};{\vec{X}}_{t-\tau_{C}:t}^{mn}|{\vec{X}}_{:t}^{\mathrm{rest}},{\vec{X}}_{:t-\tau_{C}}^{mn})}}+\underset{=D_{mn}}{\underset{︸}{I(X_{t}^{\mathrm{tar}};{\vec{X}}_{:t-\tau_{C}}^{mn}|{\vec{X}}_{:t}^{\mathrm{rest}})}}, \end{array}$$
that is,
(3b)$$\begin{array}{cc} T_{mn}= & J_{mn}+D_{mn}. \end{array}$$

Note that information flow from the two partitioned histories, $J_{mn}$ and $D_{mn}$ in Equation (Equation 3), are functions of $\tau_{C}$. Quantifying $J_{mn}$ and $D_{mn}$ along with $\tau_{C}$ allows the investigation of the memory dependency due to the evolutionary interactions of the two bundled set [10].

Partitioning ${\vec{X}}_{:t}^{mn}$ into immediate and distant causal histories further highlights the need to characterize the joint interactions of the two bundled sets in the two complementary historical states. This, again, can be achieved by using the PID approach, and is given by: (4a)$$\begin{array}{ccccc} J_{mn}= & R^{J_{}}+ & S^{J_{}}+ & U_{m}^{J_{}}+ & U_{n}^{J_{}} \end{array}$$
(4b)$$\begin{array}{ccccc} D_{mn}= & R^{D_{}}+ & S^{D_{}}+ & U_{m}^{D_{}}+ & U_{n}^{D_{}} \end{array}$$
(4c)$$\begin{array}{ccccc} T_{mn}= & R^{T_{}}+ & S^{T_{}}+ & U_{m}^{T_{}}+ & U_{n}^{T_{}}, \end{array}$$
where the last equation reflects the sum of the corresponding terms in the previous two equations. Therefore, Equation (Equation 4) illustrates the additive contribution of each information content (i.e., synergistic, redundant, and unique components) in the two partitioned histories, $J_{mn}$ and $D_{mn}$, to the entire bundled causal history, $T_{mn}$.

### 2.2. Two-Stage Dimensionality Reduction

Computing information flows in Equations (Equation 1)–(Equation 4) is infeasible due to the possibly infinite length of historical states involved, resulting in a joint probability density with infinite dimensions. To resolve this issue, we represent the temporal dependencies of the system as a Directed Acyclic Graph (DAG), where the dimensionality reduction is performed in the following two stages. First, we employ the probabilistic graphical model approach developed by Eichler and Runge [8,15] that allows a reduction of the infinite historical states in the above equations into a finite set, by assuming Markov property for the DAG. Then, a further dimension reduction is achieved through reducing the DAG by eliminating “redundant” edges. This elimination is performed by using weighted transitive reduction [16] with momentary information transfer serving as the weights. This approach is called Momentary Information Weighted Transitive Reduction (MIWTR) [12]. The second stage of dimensionality reduction is to avoid potential high albeit finite cardinalities of the joint probability in computing the information measures after the first round of reduction.

**Stage 1: From infinite to finite cardinality—a probabilistic graphical model approach.**Figure 2 illustrates the use of a DAG for time-series representation. The DAG, $G=({\vec{X}}_{:t},E)$, includes a set of directed edges *E* and a set of nodes ${\vec{X}}_{:t}$ connected by edges in *E*. Every state in ${\vec{X}}_{:t}$ is represented by a node in G, and a directed edge in *E* connecting from an earlier node $X_{t-\tau_{X}}$ to a recent node $Y_{t-\tau_{Y}}$ ($0\leq \tau_{Y}<\tau_{X}$), $X_{t-\tau_{X}}\to Y_{t-\tau_{Y}}$, refers to the direct influence from $X_{t-\tau_{X}}$ to $Y_{t-\tau_{Y}}$. An illustration of using the DAG for time-series to depict the temporal multivariate dynamics is shown in Figure 2a through a system consisting of seven components, $\left\{X_{t}^{\left(m_{1}\right)},X_{t}^{\left(m_{2}\right)},X_{t}^{\left(n_{1}\right)},X_{t}^{\left(n_{2}\right)},X_{t}^{\left(r_{1}\right)},X_{t}^{\left(r_{2}\right)},X_{t}^{\left(r_{3}\right)}\right\}$. We consider $X_{t}^{\left(r_{1}\right)}$ as the target, ${\vec{X}}_{:t}^{m}=\left\{X_{:t}^{\left(m_{1}\right)},X_{:t}^{\left(m_{2}\right)}\right\}$ as the first bundled causal history, and ${\vec{X}}_{:t}^{n}=\left\{X_{:t}^{\left(n_{1}\right)},X_{:t}^{\left(n_{2}\right)}\right\}$ as the second bundled causal history. The nodes involved in the bundled causal histories are highlighted in blue for immediate bundled causal history ${\vec{X}}_{t-\tau_{C}:t}^{mn}$ up to the partitioning time lag $\tau_{C}$, and in orange for the remaining distant bundled causal history ${\vec{X}}_{:t-\tau_{C}}^{mn}$. The historical states outsize of those two bundled sets in the system, ${\vec{X}}_{:t}^{\mathrm{rest}}=\left\{X_{t}^{\left(r_{1}\right)},X_{t}^{\left(r_{2}\right)},X_{t}^{\left(r_{3}\right)}\right\}$, are denoted as gray nodes.

Estimation of the quantities in Equations (Equation 1) and (Equation 3), and therefore the corresponding Equations (Equation 2) and (Equation 4), are challenging because the condition set has a large dimension due to its potentially very long history. Therefore, to avoid the curse of dimensionality in computing Equations (Equation 1)–(Equation 4), the Markov property for the DAG for time-series, as developed by Lauritzen et al. [17], is assumed. Loosely speaking, the Markov property for the graphical model states that a node $X_{t}$ is independent of its non-descendants in G given the knowledge of its parents, denoted by $P_{X_{t}}=\left\{Y_{t-\tau }:Y_{t}\in {\vec{X}}_{t},\tau >0,Y_{t-\tau }\to X_{t}\right\}$. By using the Markov property, the information flow from the entire ($T_{mn}$), the immediate ($J_{mn}$), and the distant ($D_{mn}$) bundled causal histories in Equations (Equation 1) and (Equation 3) can be revised as (see Figure 2b): (5a)$$\begin{array}{cc} T_{mn} & =I(X_{t}^{\mathrm{tar}};\vec{V}|{\vec{F}}_{}) \end{array}$$
(5b)$$\begin{array}{cc} J_{mn} & =I(X_{t}^{\mathrm{tar}};\vec{V}|{\vec{F}}_{},{\vec{W}}_{\tau_{C}}) \end{array}$$
(5c)$$\begin{array}{cc} D_{mn} & =I(X_{t}^{\mathrm{tar}};{\vec{W}}_{\tau_{C}}|{\vec{F}}_{}), \end{array}$$
where ${\vec{F}}_{}=P_{{\vec{X}}_{:t}^{mn}\cup X_{t}^{\mathrm{tar}}}$ is the parent set of all the nodes in the bundled causal history ${\vec{X}}_{:t}^{mn}$ and the target $X_{t}^{\mathrm{tar}}$; $\vec{V}=P_{X_{t}^{\mathrm{tar}}}\cap {\vec{X}}_{t-\tau_{C}:t}^{mn}$ is the intersection of the parents of the target, $P_{X_{t}^{\mathrm{tar}}}$, and the immediate bundled causal history, ${\vec{X}}_{t-\tau_{C}:t}^{mn}$; and ${\vec{W}}_{\tau_{C}}=P_{{\vec{X}}_{t-\tau_{C}:t}^{mn}}\cap {\vec{X}}_{:t-\tau_{C}}^{mn}$ is the parent set of the immediate history belonging to the distant history. Figure 2b illustrates $\vec{V}$, ${\vec{W}}_{\tau_{C}}$, and ${\vec{F}}_{}$ in blue, orange, and gray nodes, respectively, in the seven-component system. Equation (Equation 5) states that while information from immediate and distant bundled causal histories is aggregated at $\vec{V}$ and ${\vec{W}}_{\tau_{C}}$ influencing $X_{t}^{\mathrm{tar}}$, respectively, the conditioning on ${\vec{F}}_{}$ blocks the information from the remaining dynamics in the system, ${\vec{X}}_{:t}^{\mathrm{rest}}$, on the interaction between $X_{t}^{\mathrm{tar}}$ and ${\vec{X}}_{:t}^{mn}$.

The usage of the Markov property successfully reduces the infinite nodes in immediate (${\vec{X}}_{t-\tau_{C}:t}^{mn}$) and distant (${\vec{X}}_{:t-\tau_{C}}^{mn}$) bundled causal histories into two finite sets, $\vec{V}$ and ${\vec{W}}_{\tau_{C}}$ in Equation (Equation 5), respectively. However, the condition set ${\vec{F}}_{}$ in Equation (Equation 5) still contains possibly infinite nodes, making the computation infeasible. Therefore, we now further adopt two orders of approximations on ${\vec{F}}_{}$, and explore the corresponding implications. At the zeroth order (Order-0), we assume the condition set to be empty, i.e., ${\vec{F}}_{0}=\emptyset$. This approach of not conditioning on the states in the remaining variables ${\vec{X}}_{:t}^{\mathrm{rest}}$ allows the information from ${\vec{X}}_{:t}^{\mathrm{rest}}$ to influence the estimation of the interaction between the target $X_{t}^{\mathrm{tar}}$ and the bundled causal history. At the first order (Order-1) approximation, the condition set is allowed to include the parents of the target in the remaining variables ${\vec{X}}_{:t}^{\mathrm{rest}}$, i.e., ${\vec{F}}_{1}=P_{X_{t}^{\mathrm{tar}}}\cap {\vec{X}}_{:t}^{\mathrm{rest}}$, denoted as gray hatched nodes in Figure 2b. The Order-0 approximation mimics the idea of mutual information, which aims at capturing the shared dependency between $X_{t}^{\mathrm{tar}}$ and ${\vec{X}}_{:t}^{mn}$. On the other hand, the Order-1 approximation is consistent with the insight of transfer entropy [7], such that we prevent the influence of the states in the remaining system directly affecting the target, represented by ${\vec{F}}_{1}$, from characterizing the information flowing from the bundled causal history. Note that the simplification due to the Markov property in Equation (Equation 5) and the two approximations in the condition set ${\vec{F}}_{0}$ and ${\vec{F}}_{1}$ can be also used in computing the synergistic, redundant, and unique information in Equation (Equation 4).

**Stage 2: From high to low cardinality—MIWTR approach.** The cardinality of Equation (Equation 5) can be potentially high in a strongly interacting multivariate system, leading to higher uncertainty in the estimation of information measures. Here, we adopt a recently-proposed Momentary Information Weighted Transitive Reduction approach [12] to further reduce the dimensionality of Equation (Equation 5) by simplifying the DAG. The basic idea of MIWTR is to first exclude any “redundant” edges connecting a node in ${\vec{W}}_{\tau_{C}}$ with node in immediate history ${\vec{X}}_{t-\tau_{C}:t}^{mn}$ by using weighted transitive reduction, and then remove any node in ${\vec{W}}_{\tau_{C}}$ which are now not directly linked to the nodes in ${\vec{X}}_{t-\tau_{C}:t}^{mn}$, thereby resulting in reduced cardinality of ${\vec{W}}_{\tau_{C}}$. Here, the edge weight, representing the information flowing through the edge, is measured by momentary information transfer [8] which quantifies the shared dependency between two linked nodes conditioned on their parents. The “redundancy” of a directed edge linking two nodes by using WTR, say $X_{t-\tau_{X}}$ to $Y_{t-\tau_{Y}}$, is assessed according to the existence of an indirect path connecting $X_{t-\tau_{X}}$ and $Y_{t-\tau_{Y}}$ as well as the weights of the edges involved. That is, a directed edge, $X_{t-\tau_{X}}\to Y_{t-\tau_{Y}}$, is considered “redundant” and thus removed, if and only if there exists a path indirectly linking $X_{t-\tau_{X}}$ and $Y_{t-\tau_{Y}}$ and the minimum weight of all the edges in this indirect pathway is larger than that of $X_{t-\tau_{X}}\to Y_{t-\tau_{Y}}$. In other words, the existence of an indirect pathway, whose capacity of conveying information from $X_{t-\tau_{X}}$ to $Y_{t-\tau_{Y}}$ is stronger than the direct channel between the two nodes, makes the direct edge $X_{t-\tau_{X}}\to Y_{t-\tau_{Y}}$ obsolete. More details of MIWTR can be found in [12].

## 3. Application: Bundled Causal Interaction in Stream Chemistry Dynamics

We used this bundled causal history approach to analyze a set of published stream solute data [2] to understand how two groups consisting of cations and anions affect *pH*. The data were recorded every 7-h from March 2007 to January 2009, in the Upper Hafren catchment in the United Kingdom. The catchment, approximately 20 km from the western coast, is mainly covered by grassland over acidic soils. To investigate how different cations and anions jointly determine the *pH* level of the stream, we considered the cations {Na^+^, Al^3+^, Ca^2+^} as the first bundled set, the anions {Cl^−^, SO4^2−^} as the second bundled set, and {*pH*, lnQ (the logarithm of flow rate)} as the remaining variables. Based on the observed data shown in Figure 3a, we constructed the DAG for time-series by using Tigramite algorithm [8,18,19,20]. Generally, the algorithm first builds up preliminary links between nodes by using mutual information-based independence test, and then removes any spurious links by using CMI-based independence test by conditioning on the parents of the connected two nodes. The resulting DAG is shown in Figure 3b, with the estimation methodology for the graph detailed in [10].

Based on Equation (Equation 5), the current state of the target *pH*, the parents of the target in the bundled causal history ($\vec{V}$), and the parents of the immediate bundled causal history (${\vec{W}}_{\tau_{C}}$) are denoted in black, blue, and orange colors, respectively, in Figure 3b. The Order-1 approximation of the condition set, ${\vec{F}}_{1}$, is colored in red (note that ${\vec{F}}_{0}$ is an empty set). Figure 3b shows that ${\vec{W}}_{\tau_{C}}$ consists of 23 nodes which results in high dimensionality of the condition set ${\vec{W}}_{\tau_{C}}$. Therefore, we reduce the dimensionality of ${\vec{W}}_{\tau_{C}}$ using the MIWTR approach (see the Methodology Section for details). The reduced ${\vec{W}}_{\tau_{C}}$ obtained by using MIWTR is shown in Figure 3c, where we see that the number of nodes in ${\vec{W}}_{\tau_{C}}$ is reduced from 23 to 11.

We next computed the information flow from the entire ($T_{mn}$), immediate ($J_{mn}$), and distant ($D_{mn}$) bundled causal histories in Equation (Equation 5) as well as their synergistic, redundant, and unique components in Equation (Equation 4) by using *k*-nearest-neighbor (*k*NN) estimator [22]. *k*NN estimator was employed in this study because of its better estimation performance in using short dataset compared with other methods (e.g., binning approach and kernel density estimation [23,24]). To assess the sensitivity of choosing *k*, we computed the information flow in Equations (Equation 4) and (Equation 5) with k=[5,6,7,8,9,10,15] for both orders of approximations ${\vec{F}}_{0}$ and ${\vec{F}}_{1}$ for ${\vec{F}}_{}$. Different information contents of the PID framework in Equation (Equation 4) was estimated using a rescaled approach for calculating the redundant information proposed in [25] and also used in [9]. The results are plotted in Figure 4. The figure shows that the information measures corresponding to different *k* values generally captures similar patterns for each order of approximations. However, the limited data length (shorter than 2000) [2] and the data gaps shown in Figure 3a results in shorer usable data lengths as $\tau_{C}$ gets larger [10], and impedes reliable estimation. The limitation in data leads to the large wiggles and peaks in the estimation for some *k* values, such as the significant drop of $S^{J_{}}$ in the Order-0 approximation when k=5,6,7, and the spike in the Order-1 approximation when k=15. This outcome points to the need for further research to investigate the reliability of information metrics estimation by using *k*NN estimator under different data lengths and *k* values. Here, we chose k=10 and 5 for the Order-0 and Order-1 approximations in ${\vec{F}}_{}$, respectively, for illustration.

The estimated information components from immediate ($J_{mn}$) and distant ($D_{mn}$) bundled causal histories, over partitioning time lag $\tau_{C}$ from 5 to 150, are plotted in Figure 5a,b for the Order-0 and Order-1 approximations, respectively. In both approximations of ${\vec{F}}_{}$, the information from earlier dynamics $D_{mn}$ converges to a non-zero value with increasing $\tau_{C}$ (the area above the dotted black line). This illustrates the long-term dependence of *pH* on the selected cation and anion groups, which consist of both unique information ($U_{m}^{D_{}}$ and $U_{n}^{D_{}}$) and synergistic information ($S^{D_{}}$). Moreover, in comparing Order-0 and Order-1 approximations, while larger values of information contents are expected in Order-0 approximation (since there is no conditioning), the different portions of these information in the two approximations reveals Order-1’s condition effect. When not conditioning on the dynamics from the remaining variables (i.e., lnQ and *pH*), Figure 5a shows that cations provide very dominant unique information ($U_{m}^{J_{}}$ and $U_{m}^{D_{}}$) to the current state of *pH* in both histories. This is because cations consists of three of the overall four nodes in $\vec{V}$ in Figure 3, thus dominating the contributions that affect *pH*. Nevertheless, there still exists a certain amount of redundant information from recent dynamics, $R^{J_{}}$, and synergy from both histories, $S^{J_{}}$ and $S^{D_{}}$. This captures the overlapping and joint effects due to the dynamics of cation and anion concentrations. On the other hand, in the Order-1 approximation, given the knowledge of the historical states in the remaining variables directly affecting *pH*, that is, ${\vec{F}}_{1}=\left\{\ln Q_{t-1},{pH}_{t-1},{pH}_{t-3}\right\}$, we single out the information from the entire bundled causal history transferred only through $\vec{V}$. Preventing the impact from the remaining system on *pH*, through conditioning on ${\vec{F}}_{1}$, reduces the total information $T_{mn}$ significantly, from ∼1.3 nats to ∼0.14 nats. In particular, the redundancy in immediate bundled causal history, $R^{J_{}}$ (which is mainly induced by the dependence of solutes on flow rate [10]), diminishes. This implies no overlapping influence from cations and anions on *pH*. Meanwhile, the synergistic effect from recent dynamics, $S^{J_{}}$, now occupies a much larger proportion of the total information, illustrating the interactive influence on *pH* due to the chemical interactions between cations and anions. This analysis, which illustrates a quantitative way to characterize the information guiding the current state of the stream *pH* level transferred from the selected cations and anions in the stream, can be generalized to other multivariate systems.

## 4. Conclusions

We present an information flow-based framework to characterize the joint influence from the evolutionary dynamics of two groups of variables on the present state of a target variable. Partitioning the total information into synergistic, redundant, and unique components helps delineate different information characteristics due to the two bundled sets. This framework was applied to observed stream chemistry datasets, and successfully showed the joint impacts of cations and anions on stream *pH*.

The proposed information measures are fundamentally different from the causal detection analysis and other existing information measures (see Figure 1). The causal detection techniques (e.g., transfer entropy [7]) are used for analyzing whether and how a source affects a target, or learning a pairwise interaction pattern. Meanwhile, the proposed bundled interaction analysis, rooted in multivariate analysis, aims to characterize the joint outcome of the evolutionary dynamics of two bundled source sets. This unique feature also allows the proposed measures to be distinct from other information measures (e.g., the previous causal history analysis framework [10]).

One key issue associated with most multivariate analysis is the curse of dimensionality, which impairs reliable estimations on the corresponding measures. Here, we propose a two-stage approach to reduce the dimensions of the information measures. Based on the DAG for time-series, the Markov property is first assumed to reduce the dimensions from infinite in Equation (Equation 3) to finite in Equation (Equation 5), and a further reduction is performed by simplifying the DAG by using weighted transitive reduction. Furthermore, while the reduced graph might also affect the computed dynamics, we assume that such impact is small compared with the estimation bias induced by the high-dimensionality. This is especially true when the original graph is highly connected and many edges are removed using the two-stage approach, such as the stream chemistry example. The effectiveness of this approach was verified by the application on stream chemistry dynamics.

The proposed two orders of approximation of the influence from the rest of the system further illustrate such characterization under varying impacts from the remaining variables. Characterizing such information in both immediate and distant bundled causal histories, on the other hand, details the delineation of the influence due to a recent and the complementary earlier dynamics. In the stream chemistry application, the analysis shows that the influences of cations and anions on determining the dynamics of *pH* in the studied catchment are mainly from their synergistic effect and also from their individual impacts through unique information. Such phenomenon is masked (i.e., the top of Figure 5), when the dynamics of the remaining system (i.e., flow rate and earlier history of *pH*) are not conditioned in the computation of the information measures in Equation (Equation 5) (for the case of Order-0 approximations on ${\vec{F}}_{}$).

With the increasing availability of observed time-series data, such multivariate analysis framework opens new avenues for understanding the role of different groups of components in controlling the dynamics of a complex system.

## Figures and Tables

**Figure 1 entropy-22-00360-f001:**
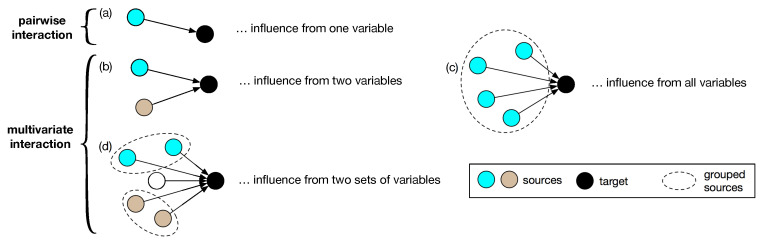
(color online) Illustration of pairwise interaction (**a**) and different types of multivariate interactions (**b**–**d**): (**a**) the influence from a source variable to a target variable; and (**b**–**d**) the influences on the target variable from two individual variables, a group of variables, and two groups of variables, respectively.

**Figure 2 entropy-22-00360-f002:**
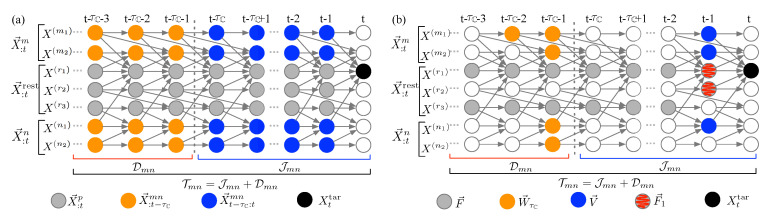
(color online) Illustration of bundled causal history analysis framework by using a system consisting of seven components: (**a**) the influence from immediate (${\vec{X}}_{t-\tau_{C}:t}^{mn}$) and distant (${\vec{X}}_{:t-\tau_{C}}^{mn}$) bundled causal histories to the target ($X_{t}^{\mathrm{tar}}$) in Equation (Equation 3); and (**b**) the different components in Equation (Equation 5) based on using the Markov property for DAG as well as the Order-1 approximation ${\vec{F}}_{1}$ for ${\vec{F}}_{}$.

**Figure 3 entropy-22-00360-f003:**
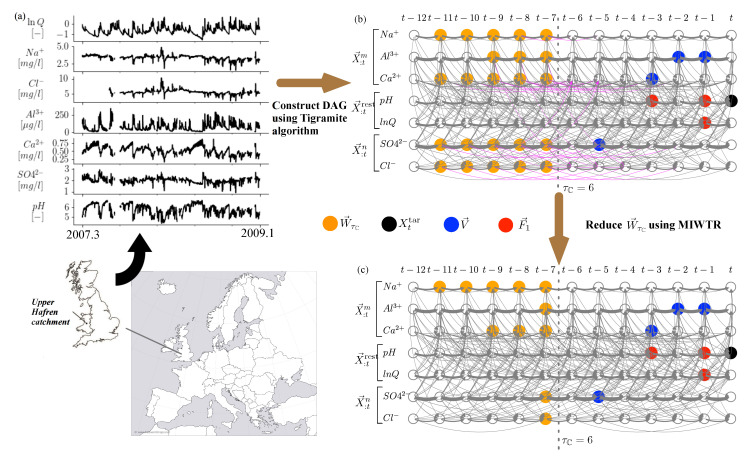
(color online) Illustration of using Momentary Information Weighted Transitive Reduction (MIWTR) to reduce the dimensionality of ${\vec{W}}_{\tau_{C}}$ for the present state of *pH* influenced by the selected cation and anion groups, with $\tau_{C}=6$. (**a**) The used stream chemistry time-series data recorded in the Upper Hafren catchment in the United Kingdom [21]. (**b**) The estimated directed acyclic graph (DAG) using Tigramite algorithm [8,18,19,20], with the original identified ${\vec{W}}_{\tau_{C}}$ (orange nodes) and $\vec{V}$ (blue nodes) in Equation (Equation 5) (the edges removed by MIWTR are highlighted in magenta). (**c**) The reduced ${\vec{W}}_{\tau_{C}}$ (orange nodes) using MIWTR.

**Figure 4 entropy-22-00360-f004:**
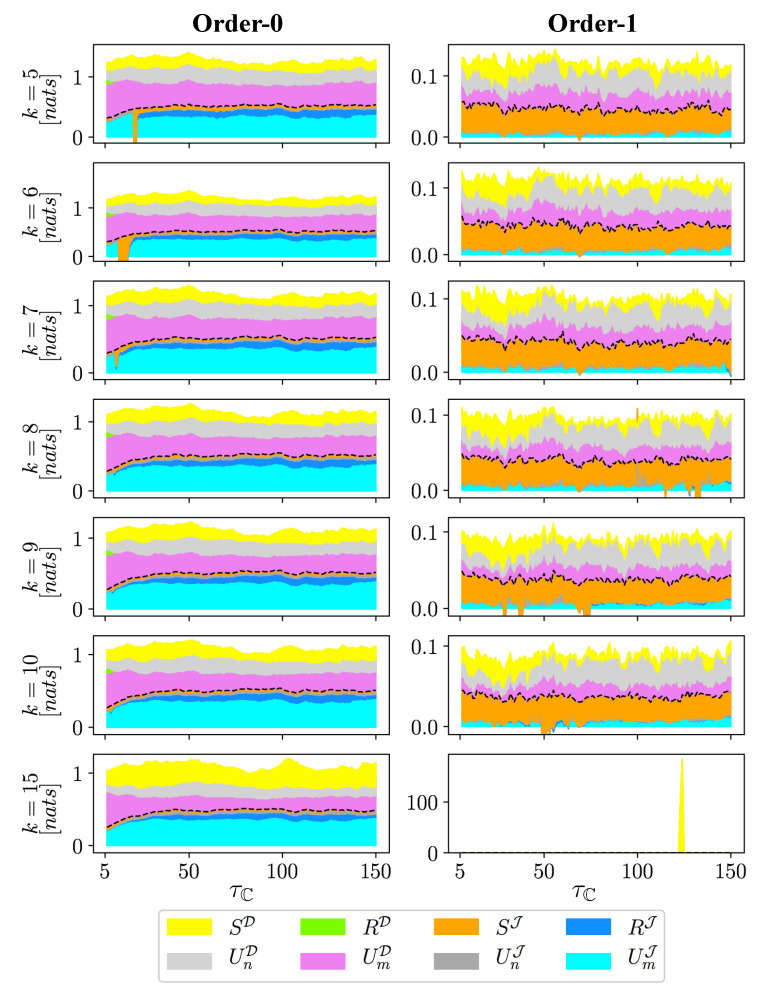
(color online) Plots of partial information decomposition of the immediate ($J_{mn}$) and distant ($D_{mn}$) bundled causal histories based on the stream solute time-series data and the estimated directed acyclic graph for time-series in Figure 2 under the Order-0 (**left**) and Order-1 (**right**) of approximation for ${\vec{F}}_{}$ by using *k*-nearest-neighbor estimators with k∈{5,6,7,8,9,10,15}.

**Figure 5 entropy-22-00360-f005:**
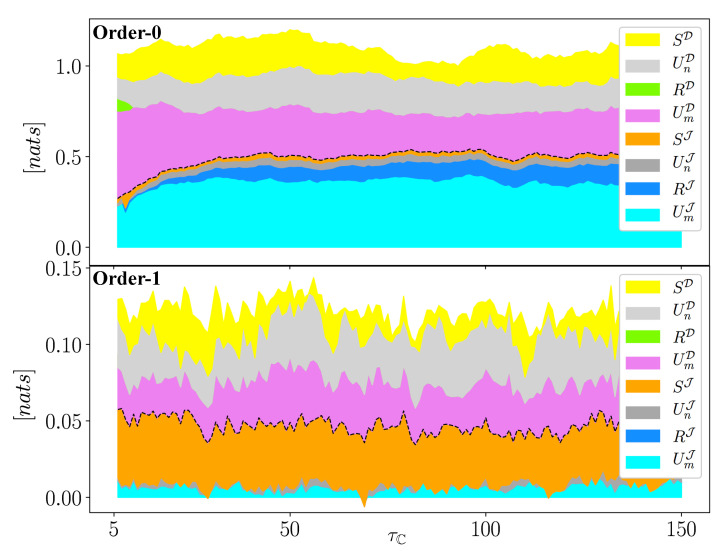
(color online) Plots of partial information decomposition of the immediate ($J_{mn}$) and distant ($D_{mn}$) bundled causal histories based on the stream solute time-series data and the estimated directed acyclic graph for time-series in Figure 3b with the Order-0 (**top**) and Order-1 (**bottom**) of approximation for ${\vec{F}}_{}$. The $J_{mn}$ and $D_{mn}$ (Equation (Equation 4a,b)) components are separated by a black dotted line.

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
