# Peer review of "Bundled Causal History Interaction"

_entropy, 2020, doi:10.3390/e22030360_

Round 1

Reviewer 1 Report

The paper "Bundled Causal History Interaction" revolves around the detection and description of causal influence from two non-overlapping subsets of variables towards the target variable.

Overall, I find the paper nicely written, with clear language, and interesting outcomes. I also appreciate the showcase of the method on stream chemistry data -- the example is sufficiently described and comprehensible even for someone who never worked or saw stream chemistry data.

I have some minor comments and questions that I believe would also help the potential future reader in making sense of your work:

  • throughout the paper, you operate with terms such as "grouped sources", "two subsets of variables", or "non-overlapping multivariate subsets"; could you elaborate on what exactly does it mean? is the definition of subsets arbitrary? problem-dependent? is there any restriction when defining subsets of variables?

Methodology:

  • some comments on how to estimate time lag τC would be helpful. Is it arbitrary? That would mean that the whole definition of immediate and distant histories is arbitrary. 
  • when reducing dimensionality in step 2 (using MIWTR approach), you remove direct edges, when there exists an indirect edge whose capacity of conveying information is stronger than the direct channel. However, indirect edges will have longer delays (time necessary) for the information to reach the target node. Wouldn't this alter the resulting dynamics that emerge from a reduced graph? Comment on this would be appreciated

Application

  • comment on why k-nearest-neighbor estimator? Have you tried using simpler binning techniques (marginal equiquantisation)? The restrictions on time series length and liability of estimate with a changing number of bins were studied before.
  • When conditioning on F1 (i.e. Order-1 approximation) the overall total information significantly drops from 1.3nats to 0.14nats -- doesn't this mean that the variables in F1 (i.e. ln Q and pH at various timesteps) have a much larger influence on the target than studied subsets (anions and cations)?
  • The comment on how synergistic effect from immediate history occupies a much larger proportion in Order-1 vs. Order-0 approximation - the much larger proportion is only relative - in absolute values it it probably smaller (not really visible in Figure 4), since total information is much lower. So maybe relatively to (very low) total information in Order-1 approximation, the SJ is large but still much smaller than in Order-0 approximation or influence of other variables (e.g. those included in F1)
  • Maybe a stupid question, but when I would compute simple transfer entropy from one subset to the target for both immediate and distant histories do I obtain the same result as using bundled histories and comparing with unique information from both histories (e.g. UIm and UDm

Minor typos, spelling, etc.:

  • "e.g. develope" line 82
  • abbreviation PID is first used on line 54 but defined at line 58
  • definition of Xn:t on line 145-146 contains three variables (n1 -- n3), however, the full system contains only n1 and n2 in subset n

Note on code availability

I very much appreciate sharing the code with the paper as authors do in the Acknowledgements. In my opinion, each computational paper should be accompanied by the code. I tried to download it and run it, however, the only possibility is to have GPU with CUDA on hand (which I do not have) since kNN estimators are written in CUDA. Can you maybe just mention this in README in your GitHub repository?

Author Response

Please find our response in the attached pdf. Thank you.

Reviewer 2 Report

The manuscript presents new information measures to quantify and characterize the interactive strength among two bundled variable sets in affecting the present state of a target variable. As the analysis of interactions among variables and especially the investigation of causal relationships between components have been applied to a wide range of systems, the paper is interesting.

I support the potential publication of this paper due to its scientific interest, but prior to that I would like the authors to clarify or fixed the following points. I recommend a minor revision of the manuscript, after which I am happy to review and provide an updated recommendation.

What is the major contribution of this work? I would like the authors to explain more about the contribution of their work in more clear way in compared with other methods.

What is the main benefit of treating variables as groups than as independent component?

I would like to ask the authors if they have used the methodology described in previous work (Jiang, P., & Kumar, P. (2019). Information transfer from causal history in complex system dynamics. Physical Review E, 99(1), 012306) . Are the results different? Please explain?

The manuscript’s introduction needs to be revised in according the previous comments.

Is there any limitation to using the methodology based on the length of data (2000 values);

How are time series grouped? Is there any methodology in the selection of time series? For example one group consists of Na+, Al3+ ,Ca2+ time series and the other consists of Cl-, SO42-time series

p. 2 Line 54. Write the explanation of PID (Partial Information Decomposition) as this abbreviation is first mentioned.

Author Response

(The authors gave the same response as above.)
